# H-DIVERGENCE: A DECISION-THEORETIC PROBABILITY DISCREPANCY MEASURE

## ABSTRACT

Measuring the discrepancy between two probability distributions is a fundamental problem in machine learning and statistics. Based on ideas from decision theory, we investigate a new class of discrepancies that are based on the optimal decision loss. Two probability distributions are different if the optimal decision loss is higher on the mixture distribution than on each individual distribution. We show that this generalizes popular notions of discrepancy measurements such as the Jensen Shannon divergence and the maximum mean discrepancy. We apply our approach to two-sample tests, which evaluates whether two sets of samples come from the same distribution. On various benchmark and real datasets, we demonstrate that tests based on our generalized notion of discrepancy is able to achieve superior test power. We also apply our approach to sample quality evaluation as an alternative to the FID score, and to understanding the effects of climate change on different social and economic activities.

## 1 INTRODUCTION

Quantifying the difference between two probability distributions is a fundamental problem in machine learning. Modelers choose different types of discrepancies, or probability divergences, to encode their prior knowledge, i.e. which aspects should be considered to evaluate the difference, and how they should be weighted. The divergences used in machine learning typically fall into two categories, integral probability metrics (IPMs, Müller (1997)), and $f$-divergences (Csiszár, 1964). IPMs, such as the Wasserstein distance, maximum mean discrepancy (MMD), are based on the idea that if two distributions are identical, any function should have the same expectation under both distributions. IPM is defined as the maximum difference in expectation for a set of functions. IPMs are used to define training objectives for generative models (Arjovsky et al., 2017), perform independence tests (Doran et al., 2014), robust optimization (Esfahani & Kuhn, 2018) among many other applications. On the other hand, $f$-divergences, such as the KL divergence and the Jensen Shannon divergence, and are based on the idea that if two distributions are identical, they assign the same likelihood to every point, so the ratio of the likelihood always equals one. One can define a distance based on the how the likelihood ratio differs from one. KL divergence underlies some of the most commonly used training objectives for both supervised and unsupervised machine learning algorithms, such as minimizing the cross entropy loss.

We propose a third category of divergences called $H$-divergences that overlaps with but does not equate the set of integral probability metrics or the set $f$-divergences. Our distance is based on a generalization (DeGroot et al., 1962) of Shannon entropy and the quadratic entropy (Burbea & Rao, 1982). Instead of measuring the best average code length of any encoding scheme (Shannon entropy), the generalized entropy can choose any loss function (rather than code length) and set of actions (rather than encoding schemes), and is defined as the best expected loss among the set of actions. In particular, given two distribution $p$ and $q$, we compare the generalized entropy of the mixture distribution $(p + q)/2$ and the generalized entropy of $p$ and $q$ individually. Intuitively, if $p$ and $q$ are different, it is more difficult to minimize expected loss under the mixture distribution $(p + q)/2$, and hence the mixture distribution should have higher generalized entropy; if $p$ and $q$ are identical, then the mixture distribution is identical to $p$ or $q$, and hence should have the same generalized entropy. We define the divergence based on the difference between entropy of the mixture distribution and the entropy of individual distributions.

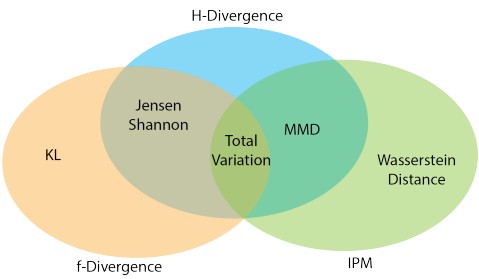

Figure 1: Relationship between H-divergence (this paper) and existing divergences. The Jensen Shannon divergence is an $f$-divergence but not an IPM; the MMD is an IPM but not an $f$-divergence; both are H-divergences. There are also H-divergences that are not $f$-divergences or IPMs.

Our distance strictly generalizes the maximum mean discrepancy and the Jensen Shannon divergence. We illustrate this via the Venn diagram in Figure 1. This generalization allows us to choose special losses and actions spaces to leverage inductive biases and machine learning models from different problem domains. For example, if we choose the generalized entropy as the maximum log likelihood of deep generative models, we are able to recover a distance that works well for distributions over high dimensional images.

To demonstrate the empirical utility of our proposed divergence, we use it for the task of two sample test, where the goal is to identify whether two sets of samples come from the same distribution or not. A test based on a probability discrepancy declares two sets of samples different if their discrepancy exceed some threshold. We use H-divergences based on generalized entropy defined by the log likelihood of off-the-shelf generative models. Compared to state-of-the-art tests based on e.g. MMD with deep kernels (Liu et al., 2020), tests based on the H-divergence achieve better test power on a large set of benchmark datasets.

As another application, we use H-divergence for sample quality evaluation, where the goal is to compare a set of samples (e.g. generated images from a GAN) with ground truth samples (e.g. real images). We show that H-divergences generally monotonically increase with the amount of corruption added to the samples (which should lead to worse sample quality), even in certain situations where the FID score (Heusel et al., 2017) is not monotonically increasing.

Finally we show that H-Divergence can be used to understand whether distribution change affect decision making. As an illustrative example, we study whether climate change affect decision making in agriculture and energy production. Traditional divergences (such as KL) let policy makers measure if the climate has changed; H-Divergence can provide additional information on whether the change is relevant to decision making for different social and economic activities.

## 2 BACKGROUND

### 2.1 PROBABILITY DISTANCES

Let $\mathcal{X}$ denote a finite set or a finite dimensional vector space, and $\mathcal{P}(\mathcal{X})$ denote the set of probability distributions on $\mathcal{X}$ that have a density. We consider the problem of defining a probability divergence between any two distributions in $\mathcal{P}(\mathcal{X})$, where a probability divergence is any function $D : \mathcal{P}(\mathcal{X}) \times \mathcal{P}(\mathcal{X}) \to \mathbb{R}$ that satisfies $D(p\|q) \geq 0, D(p\|p) = 0, \forall p, q \in \mathcal{P}(\mathcal{X})$ (Note that in general a divergence does not require $D(p\|q) > 0 \, \forall p \neq q$).

**Integral Probability Metrics** Let $\mathcal{F}$ denote some set of functions $\mathcal{X} \to \mathbb{R}$. The integral probability metrics is defined as

$$\mathrm{IPM}_{\mathcal{F}}(p\|q) = \sup_{f \in \mathcal{F}} |\mathbb{E}_p[f(X)] - \mathbb{E}_q[f(X)]|$$

Several important divergences belong to integral probability metrics. Examples include the Wasserstein distance, where $\mathcal{F}$ is the set of 1-Lipschitz functions; the total variation distance, where $\mathcal{F}$ is the set of functions $\mathcal{X} \to [-1, 1]$. The maximum mean discrepancy (MMD) (Rao, 1982; Burbea & Rao, 1984; Gretton et al., 2012) chooses a kernel function $k : \mathcal{X} \times \mathcal{X} \to \mathbb{R}_+$ and is defined by

$$\mathrm{MMD}(p\|q) = \mathbb{E}_{p,p} k(X, Y) + \mathbb{E}_{q,q} k(X, Y) - 2\mathbb{E}_{p,q} k(X, Y)$$

MMD is an IPM where $\mathcal{F}$ is the unit norm functions in the RKHS associated with the kernel $k$.

$f$**-Divergences**   Choose any convex continuous function $f : \mathbb{R}_+ \to \mathbb{R}$ such that $f(1) = 0$, the $f$-Divergence is defined as (assuming densities exist) $D_f(p\|q) = \mathbb{E}_q[f(p(X)/q(X))]$. Examples of $f$-Divergences include the KL divergence, where $f : t \mapsto t \log t$ and the Jensen Shannon divergence, where $f : t \mapsto (t+1) \log \left( \frac{2}{t+1} \right) + t \log t$.

**Scoring Rule Distance**   (Grünwald et al., 2004; Gneiting & Raftery, 2007) Another large class of probability distances are defined by proper scoring rules. A function $S : \mathcal{P}(\mathcal{X}) \times \mathcal{P}(\mathcal{X}) \to \mathbb{R}$ is called a proper scoring rule if $\forall p, q \in \mathcal{P}(\mathcal{X})$ we have $S(p,q) \geq S(p,p)$. Intuitively it is any function that is small when two distributions are identical and large when two distributions differ. Given a scoring rule $S$ we can define a distance by $D_S(p\|q) = S(p,q) - S(p,p)$.

## 2.2   H-ENTROPY

For any action space $\mathcal{A}$ and any loss function $\ell : \mathcal{X} \times \mathcal{A} \to \mathbb{R}$, the H-entropy (DeGroot et al., 1962; DeGroot, 2005; Grünwald et al., 2004) is defined as $H_\ell(p) = \inf_{a \in \mathcal{A}} \mathbb{E}_p[\ell(X, a)]$.

In words, $H$-entropy is the Bayes optimal loss of a decision maker who must select some action $a$ not for a particular $x$, but for an expectation over $p(x)$.

H-entropy generalizes several important notions of uncertainty. Examples include: *Shannon Entropy*, where $\mathcal{A}$ as the set of probabilities $\mathcal{P}(\mathcal{X})$, and $\ell(x, a) = -\log a(x)$; *Variance* where $\mathcal{A} = \mathcal{X}$, and $\ell(x, a) = \|x - a\|_2^2$; *Predictive $\mathcal{V}$-entropy*, where $\mathcal{A} \subset \mathcal{P}(\mathcal{X})$ is some subset of distributions, and $\ell(x, a) = -\log a(x)$ (Xu et al., 2020).

The most important property that we will use is that the $H$ entropy is concave.

**Lemma 1.** *(DeGroot et al., 1962) For any choice of $\ell : \mathcal{X} \times \mathcal{A} \to \mathbb{R}$, $H_\ell$ is a concave function.*

This Lemma can be proved by observing that $\inf$ is a concave function, i.e., it is always better to pick an optimal action for $p$ and $q$ separately rather than a single one for both.

$$
\begin{aligned}
H_\ell\big(\alpha p + (1 - \alpha)q\big) &= \inf_a \left( \alpha \mathbb{E}_p[\ell(X, a)] + (1 - \alpha)\mathbb{E}_q[\ell(X, a)] \right) \\
&\geq \alpha \inf_a \mathbb{E}_p[\ell(X, a)] + (1 - \alpha) \inf_a \mathbb{E}_q[\ell(X, a)] = \alpha H_\ell(p) + (1 - \alpha)H_\ell(q)
\end{aligned}
$$

This Lemma reflects why $H_\ell$ can be thought of as a measurement of entropy or uncertainty. If the distribution is more uncertain (e.g. mixture of $p$ and $q$ rather than $p$ and $q$ separately) then the optimal action always suffers a higher loss.

# 3   DEFINITION AND THEORETICAL PROPERTIES

## 3.1   H-JENSEN SHANNON DIVERGENCE

As a warm up, we first present a special case of our definition.

**Definition 1** (H-Jensen Shannon divergence)**.**

$$
D_\ell^{\mathrm{JS}}(p, q) = H_\ell \left( \frac{p + q}{2} \right) - \frac{1}{2}\big( H_\ell(p) + H_\ell(q) \big) \tag{1}
$$

The above is a divergence between $p$ and $q$ because H-entropy is concave, so $D_\ell^{\mathrm{JS}}$ is always non-negative. In particular, if we choose $H_\ell$ as the Shannon entropy, Definition 1 recovers the usual Jensen Shannon divergence. Other special choices of entropy can also recover definitions in (Burbea & Rao, 1982). In addition, we can define a divergence for any convex combination $\alpha p + (1 - \alpha)q$ where $\alpha \in (0, 1)$ but for this paper we only consider $\alpha = 1/2$.

### 3.2 GENERAL H-DIVERGENCE

In addition to the H-Jensen Shannon divergence, there are other functions based on the H-entropy that satisfy the requirements of a divergence. For example, the following quantity

$$D_\ell^{\mathrm{Min}} = H_\ell \left( \frac{p + q}{2} \right) - \min(H_\ell(p), H_\ell(q)) \tag{2}$$

is also a valid divergence (this will be proved later as a special case of Lemma 2). We can define a general set of divergences that includes the above two divergences with the following definition:

**Definition 2** (H-divergence). *For two distributions $p, q$ on $\mathcal{X}$, choose any continuous function $\phi$ : $\mathbb{R}^2 \to \mathbb{R}$ such that $\phi(\theta, \lambda) > 0$ whenever $\theta + \lambda > 0$ and $\phi(\theta, \lambda) = 0$ whenever $\theta + \lambda = 0$, define*

$$D_\ell^\phi(p\|q) = \phi \left( H_\ell \left( \frac{p + q}{2} \right) - H_\ell(p), H_\ell \left( \frac{p + q}{2} \right) - H_\ell(q) \right)$$

Intuitively $H_\ell \left( \frac{p+q}{2} \right) - H_\ell(p)$ and $H_\ell \left( \frac{p+q}{2} \right) - H_\ell(q)$ measure how much more difficult it is to minimize loss on the mixture distribution $(p+q)/2$ than on $p$ and $q$ respectively. $\phi$ is a general class of function that could convert these differences into a divergence, while satisfying the desirable properties in the next section.

The H-divergence generalizes all the previous definitions, as shown by the following proposition. Therefore any property of H-divergence is inherited by e.g. H-Jensen Shannon divergence.

**Proposition 1.** *Choose $\phi(\theta, \lambda) = \frac{\theta + \lambda}{2}$ then $D_\ell^\phi(p, q)$ is the H-Jensen Shannon divergence in Eq.(1). Choose $\phi(\theta, \lambda) = \max(\theta, \lambda)$ then $D_\ell^\phi(p, q)$ is the H-Min divergence in Eq.(2).*

### 3.3 PROPERTIES OF THE H-DIVERGENCE

We first verify that $D_\ell^\phi$ is indeed a probability divergence. In particular, Lemma 2 show that $D_\ell^\phi$ satisfies the requirements for a probability divergence. For the proof see Appendix A.

**Lemma 2.** *For any choice of $\ell$ and for any choice of $\phi$ that satisfy Definition 2, $D_\ell^\phi$ is non-negative and $D_\ell^\phi(p, q) = 0$ whenever $p = q$.*

One important property of the H-divergence is that two distributions have non-zero divergence if and only if they have different optimal actions, i.e. the optimal solutions for their respective H-entropy are different. This is shown in the following proposition (proof in Appendix A).

**Proposition 2.** $\arg\inf_a \mathbb{E}_p[\ell(X, a)] \cap \arg\inf_a \mathbb{E}_q[\ell(X, a)] = \emptyset$ *if and only if $D_\ell^\phi(p\|q) > 0$.*

Intuitively, $D_\ell^\phi$ only takes into account any difference between distributions that lead to different choice of optimal actions. This property allow us to incorporate prior knowledge about the problem. By choosing the $\mathcal{A}$ and $\ell$ we can specify which differences between distributions lead to different optimal actions, and which differences do not. The corresponding $D_\ell^\phi$ will only be non-zero when two distributions differ in way we would like to capture.

For example, we can choose $\mathcal{A}$ as a set of generative models (e.g. VAEs) and $\ell(x, a)$ is the negative log likelihood of $x$ under generative model $a$. If under two distributions we end up learning the same generative model that maximizes log likelihood, the H-divergence between them is zero.

### 3.4 RELATIONSHIP TO MMD

An important special case of the H-divergence is the set of Maximum Mean Discrepency (MMD) distances, as shown by the following theorem

**Theorem 1.** *The set of H-Jensen Shannon Divergences is strictly larger than the set of MMD distances.*

To prove this theorem for each choice of kernel $k : \mathcal{X} \times \mathcal{X} \to \mathbb{R}_+$ we construct some $\mathcal{A}$ and $\ell$ (details in Appendix A) such that

$$H_\ell \left( \frac{p+q}{2} \right) = \frac{1}{2}\mathbb{E}_p[k(X,X)] + \frac{1}{2}\mathbb{E}_q[k(X,X)] - \frac{1}{4}\mathbb{E}_{p,p}[k(X,Y)] - \frac{1}{4}\mathbb{E}_{q,q}[k(X,Y)] - \frac{1}{2}\mathbb{E}_{p,q}[k(X,Y)]$$

$$H_\ell(p) = \mathbb{E}_p[k(X,X)] - \mathbb{E}_{p,p}[k(X,Y)] \quad H_\ell(q) = \mathbb{E}_q[k(X,X)] - \mathbb{E}_{q,q}[k(X,Y)]$$

Simple algebra show that applying Definition 1 recovers the MMD distance with kernel $k$.

### 3.5 ESTIMATION AND CONVERGENCE

In many downstream tasks, we would like to estimate the H-divergence from data. Specifically we are provided with a set of i.i.d. samples $\hat{p}_m = (x_1, \cdots, x_m)$ drawn from distribution $p$ and $\hat{q}_m = (x_1', \cdots, x_m')$ drawn from distribution $q$, and would like to obtain an estimate of $D_\ell^\phi(p\|q)$ based on the samples. In this section we propose an empirical estimator for the H-divergence and show that it has nice convergence properties.

Let $\hat{D}_\ell^\phi(\hat{p}_m\|\hat{q}_m)$ be the empirical (random) estimator for $D_\ell^\phi(p\|q)$ defined by

$$\phi \left( \inf_a \frac{1}{m}\sum_{i=1}^m \ell(x_i'', a) - \inf_a \frac{1}{m}\sum_{i=1}^m \ell(x_i, a), \inf_a \frac{1}{m}\sum_{i=1}^m \ell(x_i'', a) - \inf_a \frac{1}{m}\sum_{i=1}^m \ell(x_i', a) \right)$$

where $x_i'' = x_i b_i + x_i'(1 - b_i)$ is a sample from the mixture distribution $(p+q)/2$ for $b_i$ uniformly sampled from $\{0, 1\}$.

Before presenting the convergence results, we first must define several assumptions that make convergence possible. In particular, we are going to assume that the loss function $\ell$ is $C$-bounded, i.e. there exists some $C$ such that $0 \le \ell(x, a) \le C, \forall a, x$. In addition, we assume that $\phi$ is 1-Lipschitz under the $\infty$-norm, i.e. $|\phi(\theta + d\theta, \lambda + d\lambda) - \phi(\theta, \lambda)| \le \max(d\theta, d\lambda), \forall \theta, \lambda, d\theta, d\lambda \in \mathbb{R}$ . If $\phi$ is not 1-Lipschitz we can often rescale $\phi$ to make it 1-Lipschitz. Finally, define the Radamacher complexity of $\ell$ as

$$\mathcal{R}_m^p(\ell) = \mathbb{E}_{X_i \sim p, \epsilon_i \sim \text{Uniform}(\{-1,1\})} \left[ \sup_{a \in \mathcal{A}} \frac{1}{m}\sum_{i=1}^m \epsilon_i \ell(X_i, a) \right]$$

Based on these assumptions we can bound the difference between $\hat{D}_\ell^\phi(\hat{p}_m\|\hat{q}_m)$ and $D_\ell^\phi(p\|q)$.

**Theorem 2.** *If $\ell$ is $C$-bounded, and $\phi$ is 1-Lipschitz under the $\infty$-norm, for any choice of distribution $p, q \in \mathcal{P}(\mathcal{X})$ and $t > 0$ we have*

1. $\Pr[\hat{D}_\ell^\phi(\hat{p}_m\|\hat{q}_m) \ge t] \le 4e^{-\frac{t^2 m}{2C^2}}$ *if $p = q$.*

2. $\Pr \left[ \left| \hat{D}_\ell^\phi(\hat{p}_m\|\hat{q}_m) - D_\ell^\phi(p\|q) \right| \ge 4\max(\mathcal{R}_m^p(\ell), \mathcal{R}_m^q(\ell)) + t \right] \le 4e^{-\frac{t^2 m}{2C^2}}$

For proof see Appendix A. What is most interesting about Theorem 2 is that when $p = q$, the convergence of $\hat{D}_\ell^\phi(\hat{p}_m\|\hat{q}_m)$ does not depend on the Radamacher complexity of $\ell$. This is particularly useful for two sample test, where we would like to decide if $p = q$ based on finite samples. If $p = q$ the empirical estimate $\hat{D}_\ell^\phi(\hat{p}_m\|\hat{q}_m)$ will quickly converge to 0 for relatively small sample size $m$.

## 4 EXPERIMENT: TWO SAMPLE TEST

### 4.1 TWO SAMPLE TEST

For the task of two sample test, we would like to decide if two sets of samples are drawn from the same distribution. Specifically, given two sets of samples $\hat{p}_m := (x_1, \cdots, x_m) \overset{\text{i.i.d.}}{\sim} p$ and $\hat{q}_m := (x_1', \cdots, x_m') \overset{\text{i.i.d.}}{\sim} q$ we would like to decide if $p = q$. Typically a two sample test algorithm estimates some divergence $\hat{D}(\hat{p}_m\|\hat{q}_m)$ and outputs $p \ne q$ if the divergence exceeds some threshold.

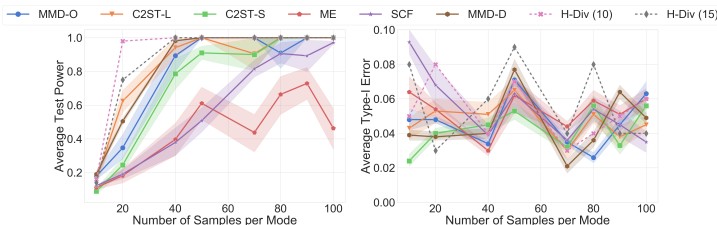

Figure 2: Average test power (**Left**) and the average type I error (**Right**) on the Blob dataset for different sample sizes. For our method (dashed line), H-Div (10) and H-Div (15) use a mixture of Gaussian distribution with 10 and 15 mixture components respectively (details in Section 4.3). Our method has significantly better test power (left plot).

There are two types of errors: *Type I error* happens when $p = q$ but the algorithm incorrectly outputs $p \neq q$. The probability that an algorithm makes a type I error is called the significance level. *Type II error* happens when $p \neq q$ but the algorithm incorrectly outputs $p = q$. The probability an algorithm does NOT make a type II error is called the *test power* (higher is better). Note that both the significance level and the test power are defined relative to the distribution $p$ and $q$.

We follow the typical setup where we would like to guarantee the significance level while empirically measuring the test power. In particular, the significance level can be guaranteed with a permutation test (Ernst et al., 2004). In a permutation test, in addition to the original set of samples $\hat{p}_m$ and $\hat{q}_m$, we also uniformly randomly swap elements between $\hat{p}_m$ and $\hat{q}_m$, and sample multiple randomly swapped datasets $(\hat{p}_m^1, \hat{q}_m^1), (\hat{p}_m^2, \hat{q}_m^2), \cdots$. The testing algorithm outputs $p \neq q$ if $\hat{D}(\hat{p}_m \| \hat{q}_m)$ is in the top $\alpha$-quantile among $\{\hat{D}(\hat{p}_m^1 \| \hat{q}_m^1), \hat{D}(\hat{p}_m^2 \| \hat{q}_m^2), \cdots\}$.[1] Permutation test guarantees the significance level (i.e. low Type I error) because if $p = q$ then swapping elements between $\hat{p}_m$ and $\hat{q}_m$ should not change its distribution, so each pair $(\hat{p}_m, \hat{q}_m), (\hat{p}_m^1, \hat{q}_m^1), \cdots$ should have the same distribution. Therefore, $\hat{D}(\hat{p}_m \| \hat{q}_m)$ happens to be in the top $\alpha$-quantile with at most $\alpha$ probability.

## 4.2 EXPERIMENT SETUP

**Dataset** We follow Liu et al. (2020) and consider four datasets: Blob (Liu et al., 2020), HDGM (Liu et al., 2020), HIGGS (Adam-Bourdarios et al., 2014) and MNIST (LeCun & Cortes, 2010).

**Baselines** We compare our proposed approach with six other divergences. All methods are based on the permutation test explained in Section 4.

- **MMD-D & MMD-O**: MMD-D (Liu et al., 2020) measures the MMD distance with a deep kernel, while MMD-O (Gretton et al., 2012) measures the MMD distance with a Gaussian kernel.
- **ME & SCF**: Mean embedding (ME) and smoothed characteristic functions (SCF) (Chwialkowski et al., 2015; Jitkrittum et al., 2016) are distances based on the difference in Gaussian kernel mean embedding at a set of optimized points, or a set of optimized frequencies.
- **C2STS-S & C2ST-L:** (Lopez-Paz & Oquab, 2017; Cheng & Cloninger, 2019) use a classifier's accuracy distinguishing between the two distributions to measure the probability discrepancy.

## 4.3 IMPLEMENTATION DETAILS

In the evaluation benchmark proposed in (Liu et al., 2020) there are two sets of samples, a training set used to tune hyper-parameters, and a validation set used to for evaluating the test performance. The training set and the validation set have the same number of samples. We also leverage the training set, but we only have one choice of hyper-parameter, which is the function $\phi$. In particular, we choose $\phi(\theta, \lambda) = \left(\frac{\theta^s + \lambda^s}{2}\right)^{1/s}$ for $s > 1$ (which includes the H-Jensen Shannon divergence when $s = 1$ and the H-Min divergence when $s = \infty$. We use the training set to find the best parameter $s$. For fair comparison each baseline method also use the training set to tune hyper-parameters as implemented in (Liu et al., 2020).

---

[1]If a tie is possible, then always break ties by sorting $\hat{D}(\hat{p}_m \| \hat{q}_m)$ in front. In this case the significance level is at most $\alpha$ (and could be less).

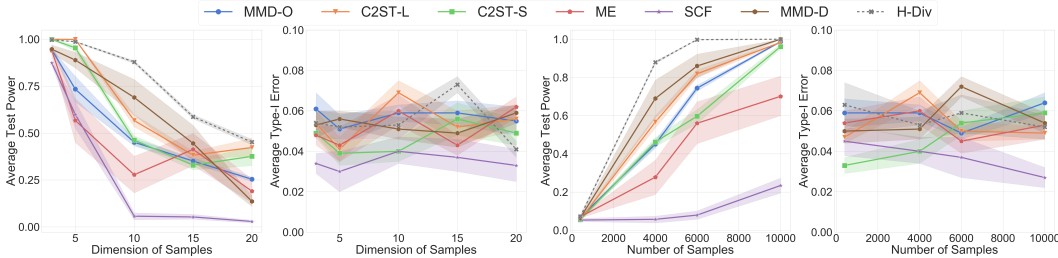

Figure 3: Average test power and type I error on HDGM dataset. **Left:** results with the same sample size (4000) and different data dimensions. **Right:** results with the same sample dimension (10) and different sample sizes. Our method (H-Div, dashed line) achieve better test power for almost every setup. In particular, when the data dimension is small, all tests have high test power. Our method scales better with data dimension and shows an advantage for higher dimensional problems.

For all the experiments, we always choose $\mathcal{A}$ to be a set of distributions, and $l(x, a)$ as the negative log likelihood of $l$ under distribution $a$. For the Blob and HDGM dataset, we choose $\mathcal{A}$ as the set of mixture of Gaussian distributions. The number of mixture component is intentionally different from the ground truth dataset to show that our approach performs well under mild mis-specification. We also provide results for Blob choosing Parzen density estimtor as $\mathcal{A}$ in Appendix B.1. For the Higgs dataset we use a Parzen density estimator, and for MNIST we use a variational autoencoder (Kingma & Welling, 2013).

Each permutation test uses 100 permutations, and we run each test 100 times to compute the test power (i.e. the percent of times it correctly outputs $p \neq q$). Finally we plot and report the performance standard deviation by repeating the entire experiment 10 times.

## 4.4 EXPERIMENT RESULTS

The average test powers are reported in Figure 2, Figure 3, Table 1 and Table 2. Our approach achieves superior test power across the board. Notably on the Higgs dataset we achieve the same test power with 2-5x fewer samples than the second best test, and on the MNIST dataset we can achieve perfect test power even on the smallest sample size evaluated in (Liu et al., 2020). Our method also produces consistent test power in comparison with the baselines as we observe much lower variance of the test power between different random draws of the dataset.

Following (Liu et al., 2020) we also evaluate the test power as the dimension of the problem increases (Figure 3). Our test power decreases gracefully as the dimension of the problem increases.

| $N$ | ME | SCF | C2ST-S | C2ST-L | MMD-O | MMD-D | H-Div |
|------|------|------|------|------|------|------|------|
| 1000 | $0.120 \pm_{0.007}$ | $0.095 \pm_{0.007}$ | $0.082 \pm_{0.015}$ | $0.097 \pm_{0.014}$ | $0.132 \pm_{0.005}$ | $0.113 \pm_{0.013}$ | $\mathbf{0.240} \pm_{\mathbf{0.020}}$ |
| 2000 | $0.165 \pm_{0.019}$ | $0.130 \pm_{0.019}$ | $0.183 \pm_{0.026}$ | $0.232 \pm_{0.032}$ | $0.291 \pm_{0.017}$ | $0.304 \pm_{0.012}$ | $\mathbf{0.380} \pm_{\mathbf{0.040}}$ |
| 3000 | $0.197 \pm_{0.012}$ | $0.142 \pm_{0.025}$ | $0.257 \pm_{0.049}$ | $0.399 \pm_{0.058}$ | $0.376 \pm_{0.022}$ | $0.403 \pm_{0.050}$ | $\mathbf{0.685} \pm_{\mathbf{0.015}}$ |
| 5000 | $0.410 \pm_{0.041}$ | $0.261 \pm_{0.044}$ | $0.592 \pm_{0.037}$ | $0.447 \pm_{0.045}$ | $0.659 \pm_{0.018}$ | $0.699 \pm_{0.047}$ | $\mathbf{1.000} \pm_{\mathbf{0.000}}$ |
| 8000 | $0.691 \pm_{0.067}$ | $0.467 \pm_{0.038}$ | $0.892 \pm_{0.029}$ | $0.878 \pm_{0.020}$ | $0.923 \pm_{0.013}$ | $0.952 \pm_{0.024}$ | $\mathbf{1.000} \pm_{\mathbf{0.000}}$ |
| 10000 | $0.786 \pm_{0.041}$ | $0.603 \pm_{0.066}$ | $0.974 \pm_{0.007}$ | $0.985 \pm_{0.005}$ | $1.000 \pm_{0.000}$ | $1.000 \pm_{0.000}$ | $\mathbf{1.000} \pm_{\mathbf{0.000}}$ |
| Avg. | 0.395 | 0.283 | 0.497 | 0.506 | 0.564 | 0.579 | **0.847** |

Table 1: Average test power $\pm$ standard error for $N$ samples over the HIGGS dataset.

| $N$ | ME | SCF | C2ST-S | C2ST-L | MMD-O | MMD-D | H-Div |
|------|------|------|------|------|------|------|------|
| 200 | $0.414 \pm_{0.050}$ | $0.107 \pm_{0.018}$ | $0.193 \pm_{0.037}$ | $0.234 \pm_{0.031}$ | $0.188 \pm_{0.010}$ | $0.555 \pm_{0.044}$ | $\mathbf{1.000} \pm_{\mathbf{0.000}}$ |
| 400 | $0.921 \pm_{0.032}$ | $0.152 \pm_{0.021}$ | $0.646 \pm_{0.039}$ | $0.706 \pm_{0.047}$ | $0.363 \pm_{0.017}$ | $0.996 \pm_{0.004}$ | $\mathbf{1.000} \pm_{\mathbf{0.000}}$ |
| 600 | $1.000 \pm_{0.000}$ | $0.294 \pm_{0.008}$ | $1.000 \pm_{0.000}$ | $0.977 \pm_{0.012}$ | $0.619 \pm_{0.021}$ | $1.000 \pm_{0.000}$ | $\mathbf{1.000} \pm_{\mathbf{0.000}}$ |
| 800 | $1.000 \pm_{0.000}$ | $0.317 \pm_{0.017}$ | $1.000 \pm_{0.000}$ | $1.000 \pm_{0.000}$ | $0.797 \pm_{0.015}$ | $1.000 \pm_{0.000}$ | $\mathbf{1.000} \pm_{\mathbf{0.000}}$ |
| 1000 | $1.000 \pm_{0.000}$ | $0.346 \pm_{0.019}$ | $1.000 \pm_{0.000}$ | $1.000 \pm_{0.000}$ | $0.894 \pm_{0.016}$ | $1.000 \pm_{0.000}$ | $\mathbf{1.000} \pm_{\mathbf{0.000}}$ |
| Avg. | 0.867 | 0.243 | 0.768 | 0.783 | 0.572 | 0.910 | **1.000** |

Table 2: Average test power $\pm$ standard error for $N$ samples over the MNIST dataset.

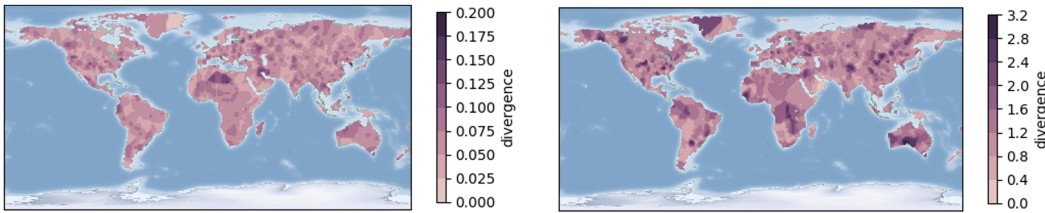

Figure 4: Example plots of H-divergence across different geographical locations for losses $\ell$ related to agriculture (left) and energy production (right). Darker color indicates larger H-divergence. Compared to divergences such as KL, H-divergence measures changes relevant to different social and economic activities (by selecting appropriate loss functions $\ell$). For example, even though climate change significantly impact the high latitude or high altitude areas, this change has less relevance to agriculture (because few agriculture activities are possible in these areas).

## 5 Experiment: Measuring Climate Change

This experiment aims to show that if a domain expert designs a loss function $\ell : \mathcal{X} \times \mathcal{A} \to \mathbb{R}$, they can obtain valuable insight from the associated H-JS divergence $D_\ell^{\mathrm{JS}}$. As an illustrative example we use climate data and study how climate change affects decision making. The H-divergence allows policy makers to quantitatively measure aspects of climate change that are relevant to decision making in different areas such as agriculture or renewable energy production.[2]

**Setup** We use the NOAA database which contains daily weather from thousands of weather stations at different geographical locations. For each location, we summarize the weather sequence of each year into a few summary statistics (the average temperature, rainfall, etc). In other words, $x \in \mathcal{X}$ represents the yearly weather summary. Let $p$ denote the empirical distribution of yearly weather for the years 1981-1999, and $q$ denote the empirical distribution of yearly weather for the years 2000-2019. $(p + q)/2$ can be interpreted as the yearly weather empirical distribution for the entire period 1981-2019. Further details of these experiments are in Appendix C.2.

**Example: Agriculture** It is known that climate changes affect crop suitability (Lobell et al., 2008). Let $\mathcal{A}$ denote the set of possible crops to plant such as wheat/barley/rice, and $\ell(x, a)$ denote the loss of planting crop $a$ if the yearly weather is $x$. We estimate the function $\ell$ by matching geographical locations in the FAO crop yield dataset (FAOSTAT et al., 2006) to weather stations in the NOAA database, and learn a function to predict crop yield from weather data with kernel ridge regression. For each geographical location we can compute the H-divergence $D_\ell^{\mathrm{JS}}$ for the estimated $\ell$ (plotted in Figure 4 left).

The H-divergence has a natural interpretation: a geographical location could either (1) plant the same crop for the entire period 1981-2019 that is optimal for the local climate (i.e. choose $a^* = \arg\min_{a \in \mathcal{A}} \mathbb{E}_{(p+q)/2}[\ell(X, a)]$); (2) plant the optimal crops for 1981-1999 and for 2000-2019 respectively. H divergence measures the additional loss of option (1) compared to option (2). In other words, it is the excess loss of not adapting crop type to climate change.

**Example: Energy production** Changes in weather also affect electricity generation, since climate change could affect the amount of wind/solar energy available. Let $\mathcal{A}$ denote the number of wind/solar/fossil fuel power plants built, and $\ell(x, a)$ denote the loss (negative utility) when the weather is $x$. We obtain the function $\ell$ by empirical estimation formulas for energy production (Npower, 2012). The H-divergence for this loss function is shown in Figure 4 (right). Intuitively the H divergence measures the excess loss of using the same energy generation infrastructure for the entire time period vs. using different infrastructure that adapts to climate change.

## 6 Experiment: Evaluating Sample Quality

We also qualitatively verify that H-Divergence defines a distance between distributions that are often consistent with human intuition. For example, this can be used to evaluate generative models, where

---

[2]Designing loss functions $\ell$ that capture the effect of climate on human activities is a well studied topic in economics and environmental science, and beyond the scope of this work. Because of the simplifications we've made, our results should be taken as an illustrative example of how domain experts might use H-divergence.

the gold standard evaluation of the distance between generated images and real images are usually with human judgement (Zhou et al., 2019). Human evaluation is expensive, so several surrogate measurements are commonly used, such as the Frechet Inception Distance (FID) (Heusel et al., 2017) or the inception score.

For this task, we choose $\mathcal{A}$ as the set of Gaussian mixture distributions on the inception feature space and $l(x, a)$ as the negative log likelihood of $x$ under distribution $a$. To evaluate the performance, we use the same setup as (Heusel et al., 2017), where we add corruption (such as Gaussian noise) from (Hendrycks & Dietterich, 2019) to a set of samples. Intuitively, adding more corruption degrades the sample quality, so a good measurement of sample quality should assign a lower quality score (higher divergence). The results are plotted in Figure 5. The remaining plots of other perturbations are in Appendix B.2. Both FID and H-divergence are generally monotonically increasing as we increase the amount of corruption. Our method is slightly better on some perturbations (such as "snow"), where the FID fails to be monotonically increasing, but our method is still monotonic.

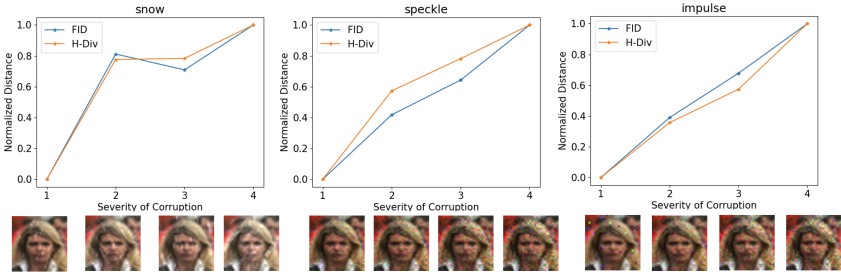

Figure 5: The divergence between corrupted image and original image measured by H-divergence vs. FID. For better comparison we normalize each distance to between $[0, 1]$ by a linear transformation. For "speckle" and "impulse" corruption, both divergences are monotonically increasing with more corruption. For "snow" corruption H-divergence is monotonic while FID is not. Other types of corruptions are provided in Appendix B.

# 7 DISCUSSION AND OPEN QUESTIONS

**Explaining improved test power**   We postulate that the test power improvements come from leveraging progress in generative model research: for each type of data (e.g. bio, image, text) there has been decades of research finding suitable generative models; we use commonly used generative models (in modern literature) for each data type (e.g. KDE for low dimensional physics/bio data, VAE for simple images). Further study is needed to further explain the observed experiment result.

**Computational efficiency**   If $\ell$ is not a convex function then evaluating the H-divergence can be computationally difficult. In particular, gradient descent does not guarantee finding $\arg\inf_a \mathbb{E}_p[\ell(X, a)]$. This can be a short-coming of H-divergence, and practitioners should interpret the empirical H-divergence estimation with caution. The practical remedy we use in our paper (when $\ell$ is non-convex) is to use the same number of gradient update steps for evaluating $H_\ell\left(\frac{p+q}{2}\right), H_\ell(p), H_\ell(q)$. Additional techniques to address non-convex optimization (such as Stein Variational Gradient Descent, restarts, beam search, etc) are interesting future work.

For H-divergence there is a trade-off between faster computation and accurate estimation of H-divergence (which should lead to improved test power). Our implementation takes about 3 hours to run on a single GPU, while the second best baseline (MMD-D) takes about 30 minutes. In time sensitive applications, studying the trade-off between computation time and test power is an interesting question.

**Statistical efficiency**   In Theorem 2 H-divergence can be efficiently estimated (i.e. independent of the Radamacher complexity of $\ell$) when $p = q$. However, MMD can be efficiently estimated even when $p \neq q$ (Gretton et al., 2012). It is an open question whether this is true for other special classes of H-divergences.

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

## A    PROOFS

**Lemma 2.** *For any choice of $\ell$ and for any choice of $\phi$ that satisfy Definition 2, $D_\ell^\phi$ is non-negative and $D_\ell^\phi(p, q) = 0$ whenever $p = q$.*

*Proof of Lemma 2.* For any choice of $p, q$ by the convexity of the H-entropy we have

$$H_\ell\left(\frac{p+q}{2}\right) - H_\ell(p) \geq \frac{1}{2}(H_\ell(q) - H_\ell(p))$$

$$H_\ell\left(\frac{p+q}{2}\right) - H_\ell(q) \geq \frac{1}{2}(H_\ell(p) - H_\ell(q))$$

Therefore we have

$$\left(H_\ell\left(\frac{p+q}{2}\right) - H_\ell(p)\right) + \left(H_\ell\left(\frac{p+q}{2}\right) - H_\ell(q)\right) \geq 0$$

By the requirement on $\phi$ we know that $\mathcal{D}_\ell^\phi(p\|q) \geq 0$.

$\square$

**Proposition 2.** $\arg\inf_a \mathbb{E}_p[\ell(X, a)] \cap \arg\inf_a \mathbb{E}_q[\ell(X, a)] = \emptyset$ *if and only if* $D_\ell^\phi(p\|q) > 0$.

*Proof of Proposition 2.* Denote $\mathcal{A}_p^* = \arg\inf_a \mathbb{E}_p[\ell(X, a)]$ and $\mathcal{A}_q^* = \arg\inf_a \mathbb{E}_q[\ell(X, a)]$. Also compute

$$H_\ell\left(\frac{p+q}{2}\right) = \inf_a \mathbb{E}_{\frac{p+q}{2}}[\ell(X, a)] = \inf_a \left(\frac{1}{2}\mathbb{E}_p[\ell(X, a)] + \frac{1}{2}\mathbb{E}_q[\ell(X, a)]\right) \tag{3}$$

If $\mathcal{A}_p^* \cap \mathcal{A}_q^* = \emptyset$, for any action $a'$ such that $\mathbb{E}_p[\ell(X, a')] = H_\ell(p)$, we must have $a' \in \mathcal{A}_p^*$ so $a' \notin \mathcal{A}_q^*$ and $\mathbb{E}_q[\ell(X, a')] > H_\ell(q)$. Similar if we choose $a''$ such that $\frac{1}{2}\mathbb{E}_q[\ell(X, a'')] = H_\ell(q)$ we have similarly have $\mathbb{E}_p[\ell(X, a'')] > H_\ell(p)$. In other words, for any choice of action $a \in \mathcal{A}$ either $\mathbb{E}_p[l(X, a)] > H_\ell(p)$ or $\mathbb{E}_q[l(X, a)] > H_\ell(q)$. Therefore

$$\inf_a \left(\frac{1}{2}\mathbb{E}_p[\ell(X, a)] + \frac{1}{2}\mathbb{E}_q[\ell(X, a)]\right) > \frac{1}{2}H_\ell(p) + \frac{1}{2}H_\ell(q) \tag{4}$$

Combining Eq.(3) and Eq.(4) we have

$$\frac{1}{2}\left(H_\ell\left(\frac{p+q}{2}\right) - H_\ell(p)\right) + \frac{1}{2}\left(H_\ell\left(\frac{p+q}{2}\right) - H_\ell(q)\right) > 0$$

By Definition 2 this would imply (for any choice of $\phi$ that satisfies the requirements in Definition 2) $D_\ell^\phi(p\|q) > 0$.

To prove the converse simply obverse that if $\mathcal{A}_p^* \cap \mathcal{A}_q^* \neq \phi$, let $a^* \in \mathcal{A}_p^* \cap \mathcal{A}_q^*$ we have $a^* = \arg\inf_{a \in \mathcal{A}} \mathbb{E}_{\frac{p+q}{2}}[l(X, a)]$. This implies that

$$2H_\ell\left(\frac{p+q}{2}\right) - H_\ell(q) - H_\ell(p) = 2\mathbb{E}_{\frac{p+q}{2}}[l(X, a^*)] - \mathbb{E}_q[l(X, a^*)] - \mathbb{E}_p[l(X, a^*)] = 0$$

By Definition 2 we can conclude that $D_\ell^\phi(p\|q) = 0$. $\square$

**Theorem 1.** *The set of H-Jensen Shannon Divergences is strictly larger than the set of MMD distances.*

*Proof of Theorem 1.* Let $k(x, y)$ be some kernel on an input space $\mathcal{X}$, and let $\mathcal{H}$ be the RKHS induced by the kernel. Define $\phi(x, y) = \|k(x, \cdot) - k(y, \cdot)\|_\mathcal{H}^2$. The MMD distance is defined by

$$\text{MMD}(p, q) = \mathbb{E}_{p,p}k(X, Y) + \mathbb{E}_{q,q}k(X, Y) - 2\mathbb{E}_{p,q}k(X, Y)$$

We can rewrite this in the following form:

$$\text{MMD}(p,q) = \mathbb{E}_{p,q}\phi(X,Y) - \frac{1}{2}\mathbb{E}_{p,p}\phi(X,Y) - \frac{1}{2}\mathbb{E}_{q,q}\phi(X,Y)$$

We also observe an algebraic relationship for any function $f(X,Y)$

$$\mathbb{E}_{\frac{p+q}{2},\frac{p+q}{2}}f(X,Y) = \frac{1}{4}\mathbb{E}_{p,p}f(X,Y) + \frac{1}{4}\mathbb{E}_{q,q}f(X,Y) + \frac{1}{2}\mathbb{E}_{p,q}f(X,Y)$$

Based on the above we can derive

$$
\begin{aligned}
\text{MMD}(p,q) &= \mathbb{E}_{p,q}\|k(X,\cdot) - k(Y,\cdot)\|_{\mathcal{H}}^2 - \frac{1}{2}\mathbb{E}_{p,p}\|k(X,\cdot) - k(Y,\cdot)\|_{\mathcal{H}}^2 - \frac{1}{2}\mathbb{E}_{q,q}\|k(X,\cdot) - k(Y,\cdot)\|_{\mathcal{H}}^2 \\
&= 2\mathbb{E}_{\frac{p+q}{2},\frac{p+q}{2}}\|k(X,\cdot) - k(Y,\cdot)\|_{\mathcal{H}}^2 - \mathbb{E}_{p,p}\|k(X,\cdot) - k(Y,\cdot)\|_{\mathcal{H}}^2 - \mathbb{E}_{q,q}\|k(X,\cdot) - k(Y,\cdot)\|_{\mathcal{H}}^2 \\
&= 4\mathbb{E}_{\frac{p+q}{2}}\|k(X,\cdot) - \mathbb{E}_{\frac{p+q}{2}}k(Y,\cdot)\|_{\mathcal{H}}^2 - 2\mathbb{E}_p\|k(X,\cdot) - \mathbb{E}_p k(Y,\cdot)\|_{\mathcal{H}}^2 - 2\mathbb{E}_q\|k(X,\cdot) - \mathbb{E}_q k(Y,\cdot)\|_{\mathcal{H}}^2 \\
&= 4\inf_{a\in\mathcal{H}}\mathbb{E}_{\frac{p+q}{2}}\|k(X,\cdot) - a\|_{\mathcal{H}}^2 - 2\inf_{a\in\mathcal{H}}\mathbb{E}_p\|k(X,\cdot) - a\|_{\mathcal{H}}^2 - 2\inf_{a\in\mathcal{H}}\mathbb{E}_q\|k(X,\cdot) - a\|_{\mathcal{H}}^2.
\end{aligned}
$$

Therefore we can define a loss $\ell : \mathcal{X} \times \mathcal{H} \to \mathbb{R}$ where

$$\ell(x,a) = 4\|k(x,\cdot) - a\|_{\mathcal{H}}^2$$

Under the new notation we have

$$
\begin{aligned}
\text{MMD}(p,q) &= \inf_a \mathbb{E}_{\frac{p+q}{2}}l(X,a) - \frac{1}{2}\left(\inf_a \mathbb{E}_p l(X,a) + \inf_a \mathbb{E}_q l(X,a)\right) \\
&= H_\ell\left(\frac{p+q}{2}\right) - \frac{1}{2}(H_\ell(p) + H_\ell(q)) = D_\ell^{\text{JS}}(p\|q)
\end{aligned}
$$

Conversely we want to show that not every H-Jensen Shannon divergence is a MMD. For example, take $H_\ell$ to be the Shannon entropy, then the corresponding $D_\ell^{\text{JS}}$ is the Jensen-Shannon divergence, which is not a MMD. This is because the JS divergence is a type of $f$-divergence, and the only $f$-divergence that is also an IPM is total variation distance. Therefore, the set of H-Jensen Shannon Divergences is strictly larger than the set of MMDs. □

**Theorem 2.** *If $\ell$ is $C$-bounded, and $\phi$ is $1$-Lipschitz under the $\infty$-norm, for any choice of distribution $p,q \in \mathcal{P}(\mathcal{X})$ and $t > 0$ we have*

1. $\Pr[\hat{D}_\ell^\phi(\hat{p}_m\|\hat{q}_m) \geq t] \leq 4e^{-\frac{t^2 m}{2C^2}}$ *if $p = q$.*

2. $\Pr\left[\left|\hat{D}_\ell^\phi(\hat{p}_m\|\hat{q}_m) - D_\ell^\phi(p\|q)\right| \geq 4\max(\mathcal{R}_m^p(\ell), \mathcal{R}_m^q(\ell)) + t\right] \leq 4e^{-\frac{t^2 m}{2C^2}}$

*Proof of Theorem 2.* Let $\hat{p}_m$ be a sequence of $n$ samples $(x_1, \cdots, x_m)$ drawn from $p$, and $\hat{q}_m$ a sequence of $n$ samples $(x_1', \cdots, x_m')$ drawn from $q$. Let $\hat{r}_m$ the sub-sampling mixture $(x_1'', \cdots, x_m'')$ defined in Section 3.5 (i.e. $x_i'' = x_i b_i + x_i'(1 - b_i)$ where $b_i$ is uniformly sampled from $\{0,1\}$). We also overload the notation $H_\ell$ by defining $H_\ell(\hat{p}_m) = \inf_{a\in\mathcal{A}} \frac{1}{m}\sum_{i=1}^m l(x_i, a)$, and define $H_\ell(\hat{q}_m), H_\ell(\hat{r}_m)$ similarly.

Before proving this theorem we need the following Lemmas

**Lemma 3.** *Under the assumptions of Theorem 2*

$$\Pr\left[H_\ell(\hat{p}_m) - \mathbb{E}[H_\ell(\hat{p}_m)] \geq t\right] \leq e^{-\frac{2t^2 m}{C^2}}$$

**Lemma 4.** *Under the assumptions of Theorem 2*

$$\Pr\left[|H_\ell(p) - H_\ell(\hat{p}_m)| \geq 2\mathcal{R}_m(\ell) + t\right] \leq e^{-\frac{2t^2 m}{C^2}}$$

To prove the first statement of the Theorem, when $p = q$ we can denote $\mu = \mathbb{E}[H_\ell(\hat{p}_m)] = \mathbb{E}[H_\ell(\hat{q}_m)] = \mathbb{E}[H_\ell(\hat{r}_m)]$ and we have

$$
\begin{aligned}
&\Pr\left[\hat{D}_\ell^\phi(\hat{p}_m \| \hat{q}_m) \geq t\right] \\
&= \Pr[\phi(H_\ell(\hat{r}_m) - H_\ell(\hat{p}_m), H_\ell(\hat{r}_m) - H_\ell(\hat{q}_m)) \geq t] && \text{Def 2} \\
&\leq \Pr\left[H_\ell(\hat{r}_m) - H_\ell(\hat{p}_m) \geq t\right] + \Pr\left[H_\ell(\hat{r}_m) - H_\ell(\hat{q}_m) \geq t\right] && \text{Def 2, Union bound} \\
&\leq \Pr\left[H_\ell(\hat{p}_m) - \mu \geq t/2\right] + 2\Pr\left[H_\ell(\hat{r}_m) - \mu \geq t/2\right] + \Pr\left[H_\ell(\hat{q}_m) - \mu \geq t/2\right] && \text{Union bound} \\
&\leq 4e^{-\frac{t^2}{2C^2/m}} && \text{Lemma 3}
\end{aligned}
$$

To prove the second statement of the Theorem, we observe that

$$
\begin{aligned}
&|\hat{D}_\ell^\phi(p_m \| q_m) - D_\ell^\phi(p \| q)| \\
&= \left|\phi\left(H_\ell(\hat{r}_m) - H_\ell(\hat{p}_m), H_\ell(\hat{r}_m) - H_\ell(\hat{q}_m)\right) - \phi\left(H_\ell\left(\frac{p+q}{2}\right) - H_\ell(p), H_\ell\left(\frac{p+q}{2}\right) - H_\ell(q)\right)\right| && \text{Def 2} \\
&\leq \max\left(\left|H_\ell(\hat{r}_m) - H_\ell(\hat{p}_m) - H_\ell\left(\frac{p+q}{2}\right) + H_\ell(p)\right|, \left|H_\ell(\hat{r}_m) - H_\ell(\hat{q}_m) - H_\ell\left(\frac{p+q}{2}\right) + H_\ell(q)\right|\right) && \phi \text{ 1-Lip} \\
&\leq \max\left(\left|H_\ell(\hat{r}_m) - H_\ell\left(\frac{p+q}{2}\right)\right| + |H_\ell(\hat{p}_m) - H_\ell(p)|, \left|H_\ell(\hat{r}_m) - H_\ell\left(\frac{p+q}{2}\right)\right| + |H_\ell(\hat{q}_m) - H_\ell(q)|\right) && \text{Jensen}
\end{aligned}
$$

Therefore, the event $|\hat{D}_\ell^\phi(p_m \| q_m) - D_\ell^\phi(p \| q)| \geq 4\max(\mathcal{R}_m^p(\ell), \mathcal{R}_m^q(\ell)) + t$ happens only if at least one of the following events happen

$$
\begin{aligned}
\left|H_\ell(\hat{r}_m) - H_\ell\left(\frac{p+q}{2}\right)\right| &\geq \mathcal{R}_m^p(\ell) + \mathcal{R}_m^q(\ell) + t/2 \geq 2\mathcal{R}_m^{(p+q)/2}(\ell) + t/2 && \mathcal{R} \text{ convex} \\
|H_\ell(\hat{p}_m) - H_\ell(p)| &\geq 2\mathcal{R}_m(\ell) + t/2 \\
|H_\ell(\hat{q}_m) - H_\ell(q)| &\geq 2\mathcal{R}_m(\ell) + t/2
\end{aligned}
$$

Based on Lemma 4 each of these events only happen with probability at most $e^{-\frac{t^2 m}{2C^2}}$. Therefore we can conclude by union bound that

$$
\Pr[|\hat{D}_\ell^\phi(p \| q) - D_\ell^\phi(p \| q)| \geq 4\max(\mathcal{R}_m^p(\ell), \mathcal{R}_m^q(\ell)) + t] \leq 4e^{-\frac{t^2 m}{2C^2}}
$$

Finally we prove the two Lemmas used in the theorem. Lemma 4 is a standard result in the Radamacher complexity literature. For a proof, see e.g. Section 26.1 in (Shalev-Shwartz & Ben-David, 2014). Lemma 3 can also be proved by standard techniques. We provide the proof here.

*Proof of Lemma 3.* Consider two sets of samples $x_1, \cdots, x_j, \cdots, x_m$ and $x'_1, \cdots, x'_j, \cdots, x'_m$ where $x_i = x'_i$ for every index $i = 1, \cdots, m$ except index $j$.

$$
\begin{aligned}
\left|\inf_a \frac{1}{m}\sum_i \ell(x_i, a) - \inf_a \frac{1}{m}\sum_i \ell(x'_i, a)\right| &\leq \sup_a \left|\frac{1}{m}\sum_i \ell(x_i, a) - \frac{1}{m}\sum_i \ell(x'_i, a)\right| \\
&= \frac{1}{m}\sup_a \left|\ell(x_j, a) - \ell(x'_j, a)\right| \leq \frac{C}{m}
\end{aligned}
$$

Then we can conclude by Mcdiarmid inequality that

$$
\Pr\left[\inf_a \frac{1}{m}\sum_i \ell(X_i, a) - \mathbb{E}\left[\inf_a \frac{1}{m}\sum_i \ell(X_i, a)\right] \geq t\right] \leq e^{-\frac{2t^2}{C^2/m}} = e^{-\frac{2t^2 m}{C^2}}
$$

$\square$

$\square$

# B    ADDITIONAL EXPERIMENTAL RESULTS

## B.1    BLOB DATASET WITH PARZEN DENSITY ESTIMATOR

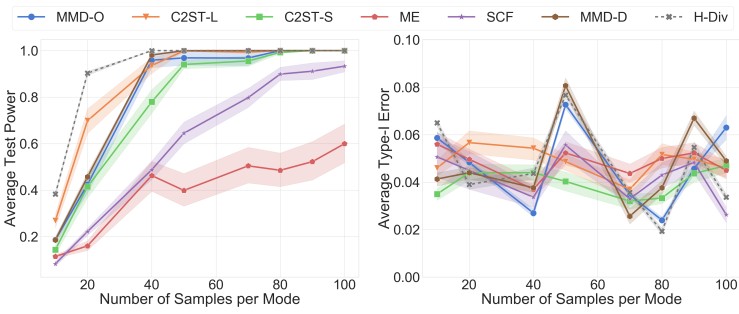

Figure 6: The same plot as Figure 2, but with Parzen density estimator and different hyper-parameters (with 30 random runs rather than 10). The conclusion is identical to Figure 2.

## B.2    COMPLETE RESULTS FOR EVALUATING SAMPLE QUALITY EXPERIMENT

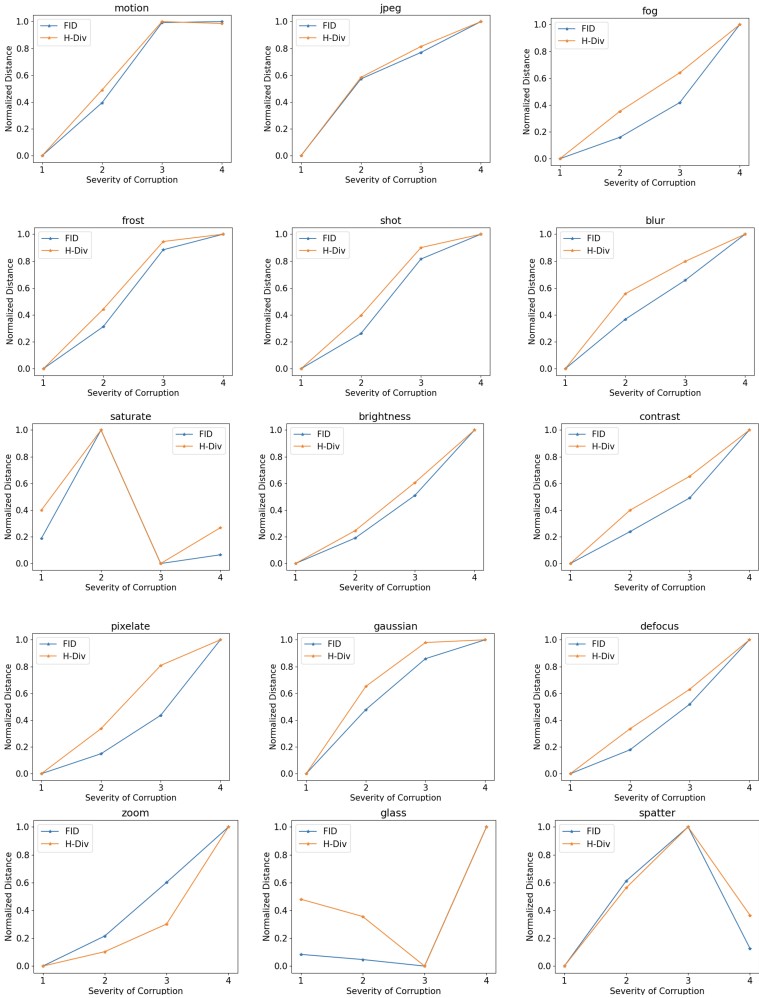

Figure 7: Additional plots that extend Figure 5.

## C    ADDITIONAL RESULTS

### C.1    CONNECTION TO OPTIMAL TRANSPORT

We first show that H-divergence can also have a transportation interpretation. For all the intuitive interpretations we avoid technical difficulty by assuming $\mathcal{X}$ is a finite set, even though all the formulas are applicable when $\mathcal{X}$ has infinite cardinality.

**Setup**    Choose $\mathcal{A} = \mathcal{X}$, and let $l(x, a)$ be a symmetric function ($l(x, a) = l(a, x)$) that denotes the cost of transporting a unit of goods from $x$ to $a$. When we say that a unit of goods is located according to $p$, we mean that there is 1 unit of goods dispersed in $\mathcal{X}$ locations, where $p(x)$ is the amount of goods at location $x$.

**Optimal Transport Distance**    The optimal transport distance is defined by

$$O_\ell(p, q) = \inf_{r_{XY}, r_X = p_X, r_Y = q_Y} \mathbb{E}_{r_{XY}}[l(X, Y)]$$

Intuitively the optimal transport distance measures the following cost: initially the goods are located according to $p$, we would like to move them to locate according to $q$; $O(p, q)$ denote the minimum cost to accomplish this transportation task.

**H-Divergence as Optimal Storage**    We first consider the intuitive interpretation to the H-entropy

$$H_\ell(p) = \inf_a \mathbb{E}_p[\ell(X, a)] \qquad a^* = \arg\inf_{a \in \mathcal{X}} \mathbb{E}_p[\ell(X, a)]$$

Suppose we want to move goods located according to $p$ to a storage location (for example, we want to collect all the mail in a city to a package center), then $a^*$ is the optimal location to build the storage location, and H-entropy measures the minimum cost to transport all goods to the storage location. Similarly $2H_\ell\left(\frac{p+q}{2}\right)$ measures the minimum cost to transport both goods located according to $p$ and goods located according to $q$ to the same storage location. The H-divergence

$$2D_\ell^{\text{JS}}(p\|q) := 2H_\ell\left(\frac{p+q}{2}\right) - (H_\ell(q) + H_\ell(p))$$

measures the reduction of transportation cost with two storage locations (one for $p$ and one for $q$) rather than a single storage location (for both $p$ and $q$).

The H-Divergence is related to the optimal transport distance by the following inequality.

**Proposition 3.** *If $\ell$ satisfies the triangle inequality $\forall x, y, z \in \mathcal{X}, l(x, y) + l(y, z) \geq l(x, z)$ then $D_\ell^{\text{JS}}(p\|q) \leq \frac{1}{2}O(p, q)$*

*Proof of Proposition 3.* Let $a_q^* = \arg\inf \mathbb{E}_q[l(X, a)]$ then we have

$$\begin{aligned}
2H_\ell\left(\frac{p+q}{2}\right) &= \inf_a \left(\mathbb{E}_p[\ell(X, a)] + \mathbb{E}_q[\ell(X, a)]\right) \leq \mathbb{E}_p[\ell(X, a_q^*)] + \mathbb{E}_q[\ell(X, a_q^*)] \\
&\leq \inf_{r_{XY}, r_X = p_X, r_Y = q_Y} \mathbb{E}_{r_{XY}}[\ell(X, a_q^*)] + \mathbb{E}_q[\ell(X, a_q^*)] \\
&\leq \inf_{r_{XY}, r_X = p_X, r_Y = q_Y} \mathbb{E}_{r_{XY}}[\ell(X, Y) + \ell(Y, a_q^*)] + \mathbb{E}_q[\ell(X, a_q^*)] \\
&= O_\ell(p, q) + 2H_\ell(q)
\end{aligned}$$

Intuitively, to move goods located according to $p$ and goods located according to $q$ to some storage location, one option is to first transport all goods from $p$ to $q$ (so that the goods at location $x$ will be $2q(x)$), then move the goods located according to $2q$ to the optimal storage location. Similarly we have

$$2H_\ell\left(\frac{p+q}{2}\right) \leq O(q, p) + 2H_\ell(p)$$

which combined we have

$$2D_\ell^{\text{JS}}(p\|q) = 2H_\ell\left(\frac{p+q}{2}\right) - (H_\ell(q) + H_\ell(p)) \leq O(p, q)$$

$\square$

### C.2 Climate Change Experiment Details

**Setup Details** In this experiment, we extract the statistics of yearly weather for each year from 1981-2019. We use the NOAA dataset, which contains daily weather data from thousands of weather stations across the globe. For each year we compute the following summary statistics: average yearly temperature, average yearly humidity, average yearly wind speed and average number of rainy days in an year. For example $x_{1990}$ is a 4 dimensional vector where each dimension correspond to one of the summary statistics above.

Let $p$ denote the uniform distribution over $\{x_{1981}, \cdots, x_{1999}\}$ and $q$ denote the uniform distribution over $\{x_{2000}, \cdots, x_{2019}\}$. For example $\mathbb{E}_p[\ell(X, a)]$ denote the expected loss of action $a$ for a random year sampled from 1981-1999. Note that for many decision problems, it is possible to make yearly decisions (e.g. decide the best crop to plant for each year). However, because we want to measure the difference between two time periods 1981-1999 vs. 2000-2019, we choose the action space $\mathcal{A}$ to be a single crop selection that will be used for the entire time period (rather than a different crop selection for each year). Similarly for energy production we choose the action space $\mathcal{A}$ to be the proportion of different energy production methods that will be used for the entire time period.

**Crop yield** We obtain the crop yield loss function $\ell(x, a)$ with the following procedure

1. We obtain the crop yield dataset from (FAOSTAT et al., 2006), each entry we extract is the following tuple: (country code, year, crop type, yield per hectare (kg/ha))

2. We associate each country code with the central coordinate (i.e. the average latitude and longitude) of the country. For each central coordinate we find the nearest weather station in the NOAA database. We use data for the nearest weather station as the weather data for the country.

3. Based on step 2 for each (country code, year) pair we can associate a weather statistics (i.e. the 4 dimensional vector described in Setup Details). We update each entry in step 1 to be (weather statistics, crop type, yield per hectare).

4. Based on the data entries we obtain in step 3 we train a Kernel Ridge regression model to learn the function $\ell(x, a)$ where $x$ is the weather statistics, $a$ is the crop type, and $\ell(x, a)$ is learned to predict the yield (normalized by market price) of the weather $x$ for crop type $a$.

**Energy production** We consider three types of energy production methods: solar, wind and traditional (such as fossil fuel). Solar energy and wind energy both depend heavily on weather, while traditional energy does not. In particular, we use empirical formulas for solar and wind energy calculation:

$$\text{solar} \propto \text{number of sunny days} * \text{daylight hour}$$

$$\text{wind} \propto \text{wind velocity}^3$$

