# OpenReview forum: "H-divergence: A Decision-Theoretic Probability Discrepancy Measure "
_ICLR.cc/2021/Conference — Reject_

### Official Review · AnonReviewer3 · 2020-10-26
**Interesting generalization of JS and MMD divergence but two-sample test part not convincing**

**Rating:** 6
**Confidence:** 4

**Review:**

This paper proposes a new class of divergences defined by a decision-theoretic perspective, in the sense that two probability distributions are different if the optimal decision loss is
higher on the mixture distribution than on each individual distribution. This new class generalizes the popular Jensen-Shannon divergence and Maximum mean discrepancy and allows to define new divergences that only take into account differences that lead to different choice of optimal actions.
Some basic properties of the new class and a theorem providing error bounds when estimating a divergence from this class from samples are shown.
A two-sample test is proposed using  a permutation approach and some experimental results are provided on standard datasets.

Pros:
- Interesting new divergence class including both JS and MMD allowing to define new divergences with a decision-theoretic perspective
- Useful theoretical results

Cons:
- I think that the decision-theoretic perspective has a great potential for making two-sample tests more meaningful but unfortunately it is not properly developed.
- Two-sample test presentation lacks rigor
- The experiments are not convincing



The paper is generally well written and the theoretical results seem correct although some proofs are a bit rushed. The experiments are not sufficiently detailed and no code is provided. The idea of generalizing divergences using a decision-theoretic perspective has already been proposed in [Grunwald&Dawid 2004] which is cited for the H-entropy but not for the generalized relative divergence, which should be mentioned. The proposed class of H divergences extending JS and MMD is novel as far as I know.
What I found especially promising but disappointingly not exploited in the two-sample test section is the following idea :
"Intuitively, $D^\phi_l$  only takes into account any difference between distributions that lead to different
choice of optimal actions. This property allow us to incorporate prior knowledge about the problem."
I would have liked to see this idea exploited in the two-sample test. Usually nonparametric two-sample tests aim at maximal power and this makes them sometimes not so useful in practice, in the sense that they summarize the output in 1 bit (saying whether p=q)  and don't tell anything about the nature of the difference, which could  be irrelevant under some decision-theoretic perspective.

I recommend rejection since the real interest of this new class is mentioned but not properly shown and the two-sample test part is not strong enough.

Detailed comments:

1) Regarding the structure: \
a) The "definition" part contains all the theoretical aspects of the proposed divergence not only the definition.\
b) The two-sample tests announced in the title only appear in the experimental part: I was expecting a section introducing definitions and literature for a general ML audience.

2) When you define a divergence in section 2.1: \
It would be useful to emphasize that your definition of divergence is broader than the usual in the sense that you can have $D(p,q) = 0$ for  $p \neq q$.

3) Regarding the choice of $\alpha=1/2$, it would be interesting to the discuss whether it would relevant or not to consider a different $\alpha$ .

4) When $D^\min_\ell$ is defined, it is said that a proof regarding its validity as a divergence is provided later but I can't find such proof.

5) After definition 2, it is claimed that $\phi$  is the most general class function…, but there is no proof.

6) Inconsistent notation between main text and proofs: \
a) $\hat{p}_m$ sometimes used as a set of samples, sometimes as an emprirical distribution which is a different object.\
b) $x''$  seems to define $\hat{r}_m$ but it is not explicitly said

7) The following statement should be more precise in what makes it "impossible" : \
" it is well known that estimating the Jensen Shannon divergence between continuous distributions is impossible with finite data."

8) You claim that part 1 (p=q) of theorem 2 is particularly useful for the two-sample test but in fact this is irrelevant in a permutation test. Under the null hypothesis (p=q), the distribution of the p-value is determined by the permutation procedure.

9) The two-sample test subsection lacks rigor. In particular:\
a) "The probability an algorithm makes a type I error is called the p-value.": This is not the standard definition of a p-value.\
b) What do you mean by a guaranteed p-value? Exact? Valid? This should be mathematically precise.\
c) In general, permutation tests should be presented with more rigor since there a different ways of doing them see e.g.
	Ernst, Michael D. "Permutation methods: a basis for exact inference." Statistical Science 19.4 (2004): 676-685.

10) Experiments:\
a) MNIST dataset: what are the two samples? Are they the same used in [Liu et al. 2020] where one of them is composed of GAN generated images?\
b) The training sizes should be explicitly specified to be able to compare to two-sample tests that compute the statistic on the whole dataset (e.g.  Hall & Tajvidi. "Permutation tests for equality of distributions in high‐dimensional settings." Biometrika 89.2 (2002): 359-374.) or those that train in an online manner (e.g. Lhéritier & Cazals. "Low-Complexity Nonparametric Bayesian Online Prediction with Universal Guarantees." NeurIPS 2019.)\
c) Figure 2: The numbers seem to correspond to the number of samples per gaussian (9), per population (2) per train/test dataset (2) (from [Liu et al. 2020]) e.g.:  20 samples in your plot means that you used 20x9x2x2=720 samples in total, if I understand correctly. \
d) I am confused by the Blobs and HDGM experiments: by using a gaussian mixture model in your test, I think that you are giving an unfair advantage to your test while at the same time penalizing it "a little bit"  with "mild" mis-specification. The aim of this setup is unclear.\
e) Figure 3: the caption should mention the name of the dataset\
f) Can you give an intuition on why your Parzen density estimation based test is superior to the other kernel methods on the HIGGS dataset? Where the additional power comes from?\
g) Overall, you should clearly justify the action space and the aim of each experiment i.e.: by choosing a specific action space, do you want to make the test more powerful than the contenders by providing it prior knowledge or do you want to tell test that only some type of differences matter (which would make it insensitive to some differences and therefore less powerful in comparison to more general tests)?

11) Proofs:\
a) Proposition 2: the proof seems to be incomplete. the last line implies that the JS is strictly positive but it doesn't seem to prove the claim for the general H-divergence.\
b) Theorem 2: sub-sampling mixture should be properly defined, I guess it corresponds to x'' in section 3.5\
c) Theorem 2: \
c1) you should clarify how you define the empirical distributions in the multivariate case \
c2) it would be helpful for the reader to explictly mention the properties used in the different derivations: 1-lipschitz, union bound, triangular inequality…\
c4) Citation of (Shalev-Shwartz & Ben-David, 2014) should be more precise\
c3) Lemma 3: First sentence: the second "consider"  should be removed I think.

Typos:\
-- diveRgence : in the abstract\
-- we are able to recover a distance that workS\
-- that ~~have~~ have a density.\
-- 100 permutationS\
-- Last line of the proof of Lemma 2: $\ell$ should appear instead of $\mathcal{V}$\
-- Proof of theorem 1 says $V$ divergence instead of $H$ Jensen Shannon Divergence\
-- Proof of theorem 2: $\hat{D}_\ell^\phi(p \Vert q)$ should be $\hat{D}_\ell^\phi(\hat{p}_m \Vert \hat{q}_m)$.

=====POST-REBUTTAL COMMENTS========

I thank the authors for the response and the efforts in the updated draft, in particular with respect to :
- the new experiment (complementing the original one on blobs) whose results are reassuring
- corrections and improvements in the proofs
- the new experiment on climate change

The two-sample test part has been modified to take into account some of my comments but still lacks rigor. It doesn't explain the permutation procedure used and whether the p-values are just valid or exact. Now, the term "significance level" (replacing "p-value") is used to refer to the probability of type I error, which is not totally accurate in general. If the p-values are just valid and not exact, the significance level is a bound on the type I error. This should be fixed in the final version.

Regarding my main concern, i.e. showing the interest of a decision-theoretic divergence using some custom loss, the new experiment on climate change is a good illustration of the interest of such divergences.

In summary, I see useful theoretical results and a step in this not-much-explored direction of tailoring divergences and two-sample tests with a decision-theoretic perspective, i.e. "detecting differences that matter", and thus I decided to raise my score.

---

> ### Author Response · Authors · 2020-11-25
> **Major Revision based on Suggestions and Answers to Specific Questions**
>
> Thank you for the incredibly detailed review. We have added a new experiment (explained in the main comment) and incorporated the writing suggestions in our updated writeup. Here we would like to answer some specific questions in your review:
>
> Q: What I found especially promising but disappointingly not exploited in the two-sample test section is the following idea : "Intuitively, $D_\ell$ only takes into account any difference between distributions that lead to different choice of optimal actions.
>
> Thank you for this suggestion. We have improved the paper with a new experiment (explained in the main comment) to explore this idea.
>
> Q: Availability of code
>
> We have released the code along with instructions to reproduce our results. To preserve anonymity we have uploaded the code to [link](https://anonymous.4open.science/r/acff7e68-1d4a-4a9f-bdc3-d551d4a395ff/).
>
> Q: You claim that part 1 (p=q) of theorem 2 is particularly useful for the two-sample test but in fact this is irrelevant in a permutation test.
>
> The significance level is guaranteed by the permutation test, but test power depends on accurate estimation of the H-divergence. In particular, H-divergence can be accurately estimated (according to Theorem 2) when $p=q$. In addition, when $p \neq q$, each of the permuted distributions $p^i, q^i, i=1, 2, \cdots$ still satisfy $p^i = q^i$. Therefore, in most situations H-divergence estimation is accurate, which could contribute to high test power.
>
> Q: Justification of choice of action space. For the Gaussian datasets our method has an unfair advantage. Where does the additional power come from?
>
> To avoid giving our method an unfair advantage, we have updated our experiment on Blobs with Parzen Density Estimation in Appendix B. We are using the same density estimator for both Blobs and Higgs datasets. The experimental results are similar.
> In addition, in section 7 we have added the following discussion: For each type of data (e.g. bio, image, text) there has been decades of research finding suitable generative models; we use commonly used generative models in modern literature for each data type (e.g. KDE for low dimensional physics/bio data, VAE for simple images). Further study is needed to further explain the observed experiment result.
>
> Q: What do you mean by a guaranteed p-value? Exact? Valid? This should be mathematically precise.
>
> These p-values are valid, we have clarified this in the experiments description.
>
> Q: For MNIST dataset: what are the two samples? Are they the same used in [Liu et al. 2020] where one of them is composed of GAN generated images?
>
> Yes, we use the test set of MNIST as one set of samples and GAN generated images as the other set of samples. This setup is identical to Liu et al, 2020 because we used their released code for this dataset.
>
> Q:The training sizes should be explicitly specified.
>
> We followed the experiment setup in  [Liu et al. 2020] and chose the training sizes to be the same as the testing sizes. We have updated our paper to clarify this.

---

### Official Review · AnonReviewer4 · 2020-10-26
**Interesting experimental results and clear presentation, but lacks motivation for the new method**

**Rating:** 5
**Confidence:** 4

**Review:**

The authors proposes a new class of divergence between probability distributions that generalizes both the Jensen-Shanon divergence and the Maximum Mean Discrepancy: the H-divergences.
Those divergences are constructed using the notion of the $H$-entropy of a probability distribution $H(p)$ defined as the minimum expected loss of an action $a$ in a set of possible actions $H(p)=\inf_{a\in A}E[l(X,a)]$ where the expectation is over samples  $X$ from $p$. Thus the $H$-entropy requires choosing a set of possible actions $A$ and a loss function $l(X,a)$.
The $H$-divergence between two probability distributions $p$ and $q$ is then obtained by comparing the difference between the $H$-entropy of the mixture distribution $(p+q)/2$ and each individual $H$-entropy of $p$ and $q$. By concavity of the $H$-entropy, the difference are both equal to $0$ only when $p$ and $q$ are equal. To allow for more generality, such differences are fed to an evaluation function $\phi$ so that the final form of the divergence is:
$D(p|q) = \phi( H((p+q)/2) - H(p), H((p+q)/2) - H(q) ) $.
The authors show that this form is general enough to include both the JS-divergence and the MMD.
An estimator of the $H$-divergence using finite samples from $p$ and $q$ is proposed along with a concentration result for this estimator that is based on the Rademacher complexity.

Experiments:
In terms of experiments, to authors consider using the new divergence for two-sample tests and evaluating sample quality.
In the two sample setting, the authors perform extensive comparisons with other SOTA methods including the recent deep kernel two sample test outperforming all those methods in terms of test power, behavior with increasing dimension of the data and sample size.

On the sample quality task, the authors show that the proposed divergence is also correlated with human judgment for images. For this task, the authors chose the set of actions to be gaussian mixture of distributions on the inception feature space and the loss $l(X,a)$ to be the likelihood of a sample $X$ under the a mixture $a$. This is shown to often better capture corruption of the samples than the FID score.






Strength:
 - The paper is clear and concise, the experiments are convincing as far as I understood.
 - The approach allows to incorporate prior knowledge about the problem by selecting a suitable set of actions $A$. In the case of two sample tests, this means that the method should capture differences between distributions that the user cares the most about. However it remains unclear if this flexibility cannot already be achieved using existing methods.

Weaknesses:
The paper does not really motivate the need for such new divergences. Hence, the impact might be limited.

The authors suggest that the additional freedom in choosing the set of actions allows to design divergences that are tailored to a particular problem. However, the authors do not explain whether this new generalization is able achieve something that already existing divergence fail to achieve. For instance, is there any particular reason to think one wouldn't achieve  similar experimental performance with a better choice of parametrization for the deep kernel? in the case of MMD-D?

In the experiments, the author make different choices for the set of actions depending on the dataset: mixture of gaussians, Parzen density estimator and for MNIST, a variational auto-encoder. However, the authors do not really explain what guides such choice.

The proposed estimator: In practice, the optimization is likely to be non-convex, especially when the set of actions is chosen to be a parametric family of probability distributions. How does this affects the estimation of the H-divergence? It doesn't seem to be an issue in the experiments, but wouldn't this be a disadvantage compared to other tests that have a closed form expression for the estimator?


Questions:
- Comparing the run time of each method? For instance SCF was designed to have a fast run time in the number of samples and tradeoff power for speed.
- On the sample quality task. How does the proposed method compare to the KID score, which has the advantage to be unbiased.

---

> ### Author Response · Authors · 2020-11-25
> **New Experiment to Address Criticism and Clarifications**
>
> Thank you for the detailed comments! In addition to a main comment that explains our major revision, we would like to answer some specific questions in your review.
>
> Q: The paper does not really motivate the need for such new divergences. Is the generalization able to achieve something existing divergence fails to achieve
>
> We have improved the paper with a new experiment (explained in our main comment) to address this criticism.
>
> Q: What guide choices of the generative models.
>
> In section 7 we have added the following discussion: We postulate that the test power improvements come from leveraging progress in generative model research: for each type of data (e.g. bio, image, text) there has been decades of research finding suitable generative models; we use commonly used generative models in modern literature for each data type (e.g. KDE for low dimensional physics/bio data, VAE for simple images). Further study is needed to further explain the observed experiment result.
>
> Q: The optimization is likely to be non-convex, how does this affect the estimation of the H-divergence?
>
> Thank you for this point, in section 7 we added the following discussion to acknowledge this short-coming and practical remedies: If $\ell$ is not a convex function then evaluating the H-divergence can be computationally difficult. In particular, gradient descent does not guarantee finding $\arg\inf_a \mathbb{E}_p[\ell(X, a)]$. Therefore, in such scenarios, practitioners should interpret the empirical H-divergence estimation with caution. The practical remedy we use in our paper (when $\ell$ is non-convex) is to use the same number of gradient update steps for evaluating $H_\ell\left( \frac{p+q}{2} \right), H_\ell(p), H_\ell(q)$. Additional techniques to address non-convex optimization (such as Stein Variational Gradient Descent, restarts, beam search, etc) are interesting future work.
>
>
> Q: Comparing the run time of each method.
>
> For typical loss functions, our run time is indeed longer than SCF or MMD, but it is within acceptable range for the dataset that we used (The slowest test across all experiments requires about 3 hours on a single CPU/GPU, while MMD-D requires 0.5 hours). We have added additional discussion to acknowledge this short-coming and propose the time-test power trade off as an open question.

---

> > ### Comment · AnonReviewer4 · 2020-11-25
> > **Thank you for your reply, but this doesn't answer my main concern**
> >
> > Hi thank you for your response.
> >
> > 1- Motivation for the method
> > The authors said a new experiment was added to motivate the proposed divergence. I'm guessing the experiment is the one in figure 4?  This figure simply shows the value of the divergence H as a heat map. I don't think one can make conclusions from this figure on the advantages of H-divegences compared other divergences. I still don't see what specific benefit it has compared to simply using the following:
> > - An MMD with deep kernel
> > - A variational lower bound based on Fenchel duality for  f-divergences such as KL. In those variational formulations, one can choose a parametric class of functions such a deep convolutional networks. This introduces an inductive bias which can be helpful for the task.
> >
> > To have a fair comparison, one should make sure to use the same inductive bias is used.  Thus the same question still stands:
> > "The authors suggest that the additional freedom in choosing the set of actions allows to design divergences that are tailored to a particular problem. However, the authors do not explain whether this new generalization is able to achieve something that already existing divergence fail to achieve. For instance, is there any particular reason to think one wouldn't achieve similar experimental performance with a better choice of parametrization for the deep kernel? in the case of MMD-D?"
> >
> >
> > 2- What guide choices of the generative models.
> > The main selling point of the paper was the ability to chose a model specifically for the task. Given that the same is possible for other divergences (f-divergence, MMD with deep kernels) and that the benefits of H-divergence remain unclear (see question before), it hard to imagine in what situation H-divergences would be more useful. If one needs to spend some engineering effort to design a suitable divergence for the tasks, can H-divergence achieve better results with less engineering? Or can the other methods achieve similar performance with the same amount of engineering?  Indeed, further study is needed to further explain the observed experiment result.
> >
> >
> > 3- Non-convexity: I think this is a serious issue with the method and as you pointed out, practitioners should interpret the results with caution. This is especially the case since more flexible models often result in non-convex losses. Other methods do not always have this limitation, for instance optimizing the parameter of a kernel still yields a valid MMD even when only local minima are obtained for the parameters.
> >
> > While solving non-convex problems is indeed a hard problem with many open theoretical questions, the point here is that the proposed method can be very sensitive to this optimization problem. This is not necessarily the case for other methods and since the paper doesn't really explain the advantages  over those (see point 1), I find this very concerning.

---

### Official Review · AnonReviewer2 · 2020-11-01
**Very Interesting and Significantly Novel**

**Rating:** 9
**Confidence:** 4

**Review:**

This paper proposes a H divergence that is a generalization of many popular f divergences and IPMs. The paper gives an empirical estimator with convergence rates for this divergence, where the rates are very fast when the two distributions are equal. The paper shows how the empirical estimator has practical use for two sample tests and measuring the corruption of a sample. The proposed H divergence is "useful" when the two distributions are close to each other, but as the authors acknowledge in the future work, it is an open question whether it could be "useful" in other cases.

Overall I think this paper is very interesting and has a lot of novelty. I am not extremely on top of the most recent literature on measuring differences between probability distributions, so there may be literature that is not being reviewed and ignored, but from an "outsiders" perspective this seems to be a significant contribution to the area. There are some minor grammar issues (see minor comments for the ones I caught) and the paper could use a thorough re-read for grammar in general.

Major Comments:
The proof for Proposition 2 shows that if the intersection between the optimal action spaces of p and q is empty, then the divergence is greater than 0. However it is not seem obvious that the converse is true i.e. if the divergence is greater than 0 then the intersection is empty. If the converse is trivial, having some explanation for it would be helpful.

The notation for the proof of Lemma 3 is rather confusing. The authors want to state that the two samples are equal except for at the points x_j and x'_j, but writing "i \neq j" seems to imply at first glance that the two samples are not equal where the indexes don't line up instead i.e. x_1 \neq x'_2. It would be clearer to just state x_i = x'_i except at one index j. Also remove the second "consider", it is not necessary.

Is it computationally possible to run the experiments more than 10 times? The power looks good but the type I errors still look a little noisy. Granted the scale is very small, but 10 is not considered a large number of "simulations".

Minor Comments:
- Please make Figures 2 and 3 bigger. There seems to be some white space you can play with and there is some room before the 8 page limit.

- Need an s here: "distance that work(s) well for distributions over high dimensional images."

- Remove "among of" here: "We show that H-divergences generally monotonically increases with the among of corruption added to the samples "?

- Use "or" instead of a slash here: "entropy / uncertainty"

- Remove "in order" here: "measure how much more difficult it is in order to minimize loss on the mixture distribution"

- Insert "that" here: "probability (that) an algorithm makes a type"

- "test power" is a strange term. Normally it is referred to as just "power" or "statistical power" or "power of a test"

- The caption in Figure 3 has (Left 2) and (Right 2) which are weirdly bolded and might be better written as (two on the left) and (two on the right)? Also missing an "s" here: "Our method (H-Div, dashed line) achieve(s)"

- Missing "s" here: "Each permutation test use(s) 100 permutation(s)"

---

> ### Author Response · Authors · 2020-11-25
> **Thank you for the suggestions**
>
> Thank you for your review! In addition to a main comment that explains our major revision, we would like to answer some specific questions.
>
> Q: The proof for Proposition 2 is incomplete
>
> We have updated the paper with a detailed step-by-step proof.
>
> Q: The notation for the proof of Lemma 3 is rather confusing.
>
> We have updated the paper with the suggested revision.
>
> Q: The power looks good but the type I errors still look a little noisy.
>
> Thank you for this suggestion, we used exactly the experiment setup of Liu et al, 2020 for accurate comparisons. In the revised paper, we also plot Figure 2 with additional runs in Appendix B. The results are less noisy but the conclusions do not change.

---

### Official Review · AnonReviewer5 · 2020-11-04
**A new divergence for two sample tests**

**Rating:** 7
**Confidence:** 5

**Review:**

Summary: The distance or divergence between two probability distributions is essential for machine learning. This paper introduces a new class of divergence functions based on optimal decision loss function. They first introduce a class of entropy functional, namely the loss function depending on the action and state. This type of function extends the classical entropy function, including negative Boltzman-Shannon entropy. Using it, they further construct a divergence based on the mixture of probability densities.  Several propositions and numerical experiments demonstrate the effectiveness of proposed divergence functions.

Pro: The idea is clean, and the derivation is correct. Several analytical examples and detailed comparisons are presented.

I still have some questions about the proposed approach. I may increase the rating if the author can address them.

1. Have the author discuss the related Hessian metric for the proposed divergence function? For the KL divergence or Jen-Shannon divergence, the Hessian matrix for D(p, p+dp) recovers the Fisher information matrix. For the proposed approach, what is the corresponding information matrix?

See related studies in

Li, Zhao, Wasserstein information matrix, 2019.

2. The authors claim this divergence contains part of IPM (integral probability metrics). It would be better for the authors to discuss the relation of this proposed divergence with Wasserstein type divergences functions.

3. Does the author consider the duality for the proposed divergence functionals? This could be useful in generative models.

---

> ### Author Response · Authors · 2020-11-25
> **New revision with Suggested Theory**
>
> Thank you for the suggestions! In addition to a main comment that explains our major revision, we would like to answer some specific questions in your review.
>
> Q: The related Hessian metric for the proposed divergence function?
>
> H-divergence induces a Hessian metric; studying the metric and its applications was intended as a follow up work. Here is a preview of the results:
> Given some parametric model $p_\theta$ and assume $\ell$ is twice differentiable, we have $D_\ell^{\mathrm{JS}}(p_\theta \Vert p_{\theta + d\theta}) = \frac{1}{8} d\theta^T G_\theta d \theta + o(\lVert d\theta \rVert_2^2)$ where
>
> $$
> G_\theta = \mathbb{E}_\theta[\nabla_a l(X, a^*) \nabla_\theta \log p_\theta(X)^T]^T \mathbb{E}_\theta[\nabla^2_a l(X, a^*)]^{-1} \mathbb{E}_\theta[\nabla_a l(X, a^*) \nabla_\theta \log p_\theta(X)^T]
> $$
>
> and $a^* = \arg\inf_{a} \mathbb{E}_{p_\theta}[\ell(X, a)]$. In the special case where $D^\ell_\mathrm{JS}$ is the Jensen Shannon divergence, we also recover the Fisher matrix.
>
> Q: Relation of this proposed divergence with Wasserstein type divergences functions.
>
> Our divergence also defines a type of transport distance --- though different from standard Wasserstein type distances. We added additional discussion and theoretical results in Appendix C.
> As a highlight of the additional result: Let $\mathcal{A} = \mathcal{X}$ and let $\ell(x, a)$ be a symmetric function that denotes the cost of transporting one unit of goods from location $x$ to location $a$. Let $O_\ell(p, q)$ be the optimal transport distance (with transportation cost $\ell$), the key result is the following inequality $D_\ell^{\mathrm{JS}}(p \Vert q) \leq \frac{1}{2} O_\ell(p, q)$ whenever $\ell$ is a proper distance (i.e. it satisfies symmetry and triangle ineq). We further note that H-divergence here is zero if and only if $p = q$, so it is a non-vacuous lower bound to the optimal transport distance.
>
> Q: Consider the duality for the proposed divergence functionals and its use in generative models.
>
> Thank you for the suggestion. We are actively working on using H-divergence to train generative models as a follow up work.

---

### Author Response · Authors · 2020-11-25
**Major Revision with New Experiments and Theory based on Reviewer Suggestions**

We thank all reviewers for their detailed reviews and great suggestions on improving our paper. All reviewers commented positively on the novelty of the proposed divergence and its generalization of MMD and JSD. Major concerns include 1. further motivation for the proposed divergence 2. rigor of experiments 3. open theoretical questions.

We have made significant revisions to improve the paper according to reviewer suggestions. Notable changes include:

New experiment (R3, R4) in Section 5: We use H-divergence to measure how changes in climate affect economic decisions. By designing a suitable decision loss function, H-divergence measures aspects of climate change that are relevant to decision making in different areas such as agriculture or energy production. As concrete examples, we use historical weather data and empirical models relating weather to crop yields and energy production. We can then use H-divergence to measure how historical changes in weather (climate change) affect optimal crop selection strategies and energy portfolio selection strategies. We plot these 2 measures for thousands of different locations across the globe (Figure 4).

New theoretical results (R5): We derive a closed form solution for the associated Hessian metric. We also show that the H-divergence family includes distances with optimal transportation interpretations (Appendix C.1).

Minor changes to writing or experiment rigor are explained in each reviewer’s comment section.

---

### Decision · Program_Chairs · 2021-01-07
**Final Decision**

**Decision:**

Reject

**Comment:**

The focus of the submission is the measuring of the discrepancy of two probability distributions. Using the notion of H-entropy, the authors propose a new divergence measure the H-divergence (Def. 2), as a common generalization of Jensen-Shannon divergence and maximum mean discrepancy. They suggest an empirical estimator for H-divergence and show that it is consistent (Theorem 2). The efficiency of the technique is illustrated in the context of 2-sample testing, sample quality evaluation and measuring climate change.

Overall, the submission addresses an important problem (defining a class of divergence measure). As pointed out by the reviewers, however fundamental questions on the proposed estimator are not addressed:
1)the motivation of using a set of actions (in the definition of H-divergence) and how to design them are unclear,
2)the impact of the loss function l (in the definition of H-divergence) is not explored: this can easily lead to instabilities due to the lack of closed-form solution and by the arising non-convex optimization task.
The gain compared to existing solutions have to be understood and explored further.